# DeFine: Enhancing LLM Decision-Making with Factor Profiles and Analogical Reasoning

## Abstract

LLMs are ideal for decision-making due to their ability to reason over long contexts and identify critical factors. However, challenges arise when processing transcripts of spoken speech describing complex scenarios. These transcripts often contain ungrammatical or incomplete sentences, repetitions, hedging, and vagueness. For example, during a company's earnings call, an executive might project a positive revenue outlook to reassure investors, despite significant uncertainty regarding future earnings. It is crucial for LLMs to incorporate this uncertainty systematically when making decisions. In this paper, we introduce DeFine, a new framework that constructs probabilistic factor profiles from complex scenarios. DeFine then integrates these profiles with analogical reasoning, leveraging insights from similar past experiences to guide LLMs in making critical decisions in novel situations. Our framework separates the tasks of quantifying uncertainty in complex scenarios and incorporating it into LLM decision-making. This approach is particularly useful in fields such as medical consultations, negotiations, and political debates, where making decisions under uncertainty is vital.

## 1 Introduction

Large language models are increasingly utilized for decision-making, thanks to their advanced reasoning abilities (Eigner & Händler, 2024). While research has examined various types of reasoning, e.g., deductive, inductive, mathematical, and multi-hop reasoning, most studies have tackled simpler tasks, such as natural language inference and math word problems (Bostrom et al., 2022; Huang & Chang, 2023; Sprague et al., 2024; Mondorf & Plank, 2024). There is a significant gap in handling complex, real-world scenarios, such as making financial investment decisions (Keith & Stent, 2019), where the stakes are high and poor decisions can result in severe consequences. Therefore, it is crucial to understand how LLMs make decisions, allowing domain experts to collaborate with them to make informed, rational decisions in complex situations.

The challenges are compounded when LLMs are required to handle long contexts, extract multiple relevant pieces of information, and make decisions based on this data (Krishna et al., 2023; Laban et al., 2024). Key issues include a tendency to prioritize information at the beginning and end of the context (recency bias; Liu et al. 2023), handling inconsistencies, and mitigating hallucinations in numerical data (Hu et al., 2024a;b). Current tools such as chain-of-thought (CoT; Wei et al. 2023), tree-of-thought (ToT; Yao et al. 2023), Reflexion (Shinn et al., 2023), are designed to provide reasoning traces for LLM decisions; however, their explanations remain ambiguous. They lack ***precise, quantitative insights into key factors and the degree of uncertainty involved***. Consequently, decision-makers are left with doubts about the reliability of these decisions and how they can be improved. There is a pressing need to enhance the verifiability of LLMs in complex decision-making scenarios to ensure their dependability and effectiveness.

We present DeFine, a new framework designed to build probabilistic factor profiles from transcripts of spoken speech that describe complex scenarios. These transcripts are often excessively long, containing ungrammatical sentences, repetitions, hedging, and vagueness (Sawhney et al., 2020; Medya et al., 2022). For example, during a quarterly earnings call, a company executive might project a positive revenue outlook to boost investor confidence, despite significant uncertainties surrounding these projections (Mukherjee et al., 2022). DeFine constructs a factor profile for each transcript that summarizes essential information into a set of factors and estimates the probabilities of potential outcomes for these factors. Moreover, we employ the Bradley-Terry model (Bradley & Terry, 1952) to identify dominant factors and evaluate how these factors collectively impact decision-making. Our

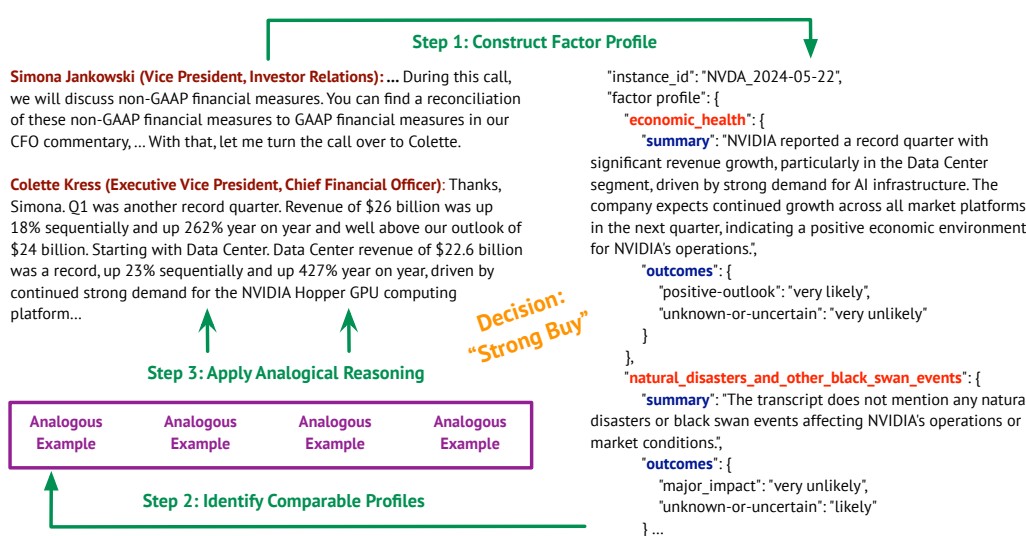

Figure 1: An excerpt from a typical earnings call transcript and its associated factor profile.

factor profiles are designed to capture the nuances in spoken transcripts. This includes not just what is explicitly stated, but also the implications of details that are omitted, providing a viable method for quantifying uncertainty.

***Our research integrates probabilistic factor profiles with analogical reasoning***, a type of reasoning that identifies connections between similar situations to facilitate knowledge transfer from a familiar context to a novel one (Webb et al., 2023; Yasunaga et al., 2024). It helps decision-makers draw parallels between current situations and past experiences, thus effectively leveraging historical insights to inform their decisions. Instead of relying on traditional text matching, we use factor profiles to retrieve analogous examples, which identifies historical cases with similar levels of uncertainty across key dimensions. Analogical reasoning further sets our work apart from traditional Bayesian inference frameworks used in decision-making (Halawi et al., 2024; Lin et al., 2024; Liu et al., 2024). The latter often require extensive sampling during inference, which tends to increase inference costs and potentially leads to latency issues. The contributions of our research are summarized as follows.

- We introduce DEFINE, a new framework designed to enhance decision-making in LLMs. DEFINE utilizes probabilistic factor profiles to quantify uncertainty in complex scenarios, along with analogical reasoning that leverages insights from similar past experiences to guide LLMs in making crucial decisions in novel situations. Our framework aims to boost the utility of LLMs by adding a layer of transparency to their decision-making processes.

- Our research has produced actionable insights for predicting stock movement trends by analyzing earnings call transcripts. These insights enable investors to make informed, data-driven decisions. Furthermore, our approach has potential applications beyond finance, including in fields such as medical consultations and political debates (Lehman et al., 2022), where the discussions involve complex issues and the decisions made can have significant consequences.[1]

## 2 OVERVIEW AND FACTOR ANALYSIS

LLMs have shown great promise in financial services (Reddy et al., 2024; Lee et al., 2024), yet the specifics of their decision-making processes remain largely unexplored. In this paper, we investigate ***how LLMs can guide investment decisions by analyzing earnings call transcripts***. The transcripts contain a rich mix of textual and numerical data, presenting a unique challenge for LLMs in making rational decisions. An earnings call is a teleconference in which the management of a public company discusses its financial results with analysts and investors for a specific reporting period, such as a quarter or a fiscal year.[2] The transcript generally consists of two sections: *the initial prepared remarks from the company's executives*, and *a subsequent Q&A session*. During these calls, executives provide

---

[1]We plan to make the data and code publicly available upon acceptance to facilitate research in this area.

[2]https://en.wikipedia.org/wiki/Earnings_call

a deep dive into the company's financials, discuss key performance indicators, and share strategic plans for the future. An excerpt from a typical earnings call transcript is shown in Figure 1.

Our goal is to provide investment recommendations based on earnings call transcripts (ECTs), using a five-tier system: *Strong Buy*, *Buy*, *Hold*, *Sell*, and *Strong Sell*. We choose this approach over a simple binary classification (Ni et al., 2024) to give a clearer and more nuanced assessment of the investment opportunity. We highlight the key drivers in decision-making by identifying a small set of factors from a lengthy transcript (Eigner & Händler, 2024; Feng et al., 2024). For example, in an earnings call, discussions of financials such as revenue, expenses, and profit margins can be overwhelming. A factor profile helps to distill these discussions into multiple variables, effectively reducing information redundancy and allowing decision-makers to focus on the most impactful factors. Crucially, a factor profile offers a comprehensive view on an earnings call. If critical elements such as debt levels are not addressed by company executives, they can be marked as 'unknown or uncertain.' This contrasts with textual summaries of the transcript (Cho et al., 2021; Khatuya et al., 2024), which may be biased toward the topics emphasized by executives and discussed during the Q&A session.

## 2.1 FACTOR PROFILE

Let $X$ denote an earnings call transcript, based on which we predict a stock investment decision $Y$, which can take one of 5 categorical outcomes: $\{\text{strong buy}, \text{buy}, \text{hold}, \text{sell}, \text{strong sell}\}$. We construct a factor profile for each transcript $X$. Specifically, we define a set of factors $\mathcal{F} = \{F_1, F_2, \ldots, F_n\}$, where each factor $F_i$ is associated with multiple potential outcomes $O_{i1}, O_{i2}, \ldots, O_{im}$. The likelihood of each outcome, given the transcript, is modeled by a probabilistic function, $P(O_{ij}|X)$. These probabilities are inferred using a methodology that optimally integrates textual reasoning with quantitative analysis. Thus, each factor outcome's probability informs the aggregation model that predicts the investment decision $Y$.

In this study, we focus on a curated set of 15 factors, categorized into three groups: *macroeconomic influences* (e.g., economic health, market sentiment), *company-specific dynamics* (e.g., mergers and major acquisitions, product launches), and *historical financial metrics* (e.g., past earnings, stock prices). These factors were carefully selected through an iterative process of querying the LLM for key variables crucial in forecasting stock movements following earnings announcements. We intentionally limited our variable set to 15 factors, each with two to three potential outcomes, as detailed below. By distilling the analysis to a few significant predictors, our approach balances complexity and performance, while also allowing for future integration of domain-specific factors identified by financial analysts.

| |
|---|
| 1. Economic Health |
| 2. Market Sentiment and Investor Psychology |
| 3. Political Events and Government Policies |
| 4. Natural Disasters and Black Swan Events |
| 5. Geopolitical Issues |
| 6. Mergers and Major Acquisitions |
| 7. Regulatory Changes and Legal Issues |
| 8. Financial Health |
| 9. Company Growth |
| 10. Company Product Launches |
| 11. Supply Chain |
| 12. Technological Innovation |
| 13. Historical Earnings Per Share (EPS) |
| 14. Historical Revenue |
| 15. Historical Stock Prices |

Table 1: A curated set of 15 factors for forecasting stock movements following earnings.

- **Macroeconomic Influences.** These encompass broad economic factors that affect the entire market or large segments of it. This includes the overall economic health, market sentiment, political events, natural disasters and geopolitical issues (Liu et al., 2024). Each factor leads to two potential outcomes; for instance, natural disasters might cause a 'Major Impact' by disrupting economies and global supply chains, and directly affecting market performance; the 'Unknown or Uncertain' outcome reflects the unpredictability of such events.

- **Company-Specific Dynamics.** These factors are linked to the internal operations and strategic decisions of individual companies, such as mergers and acquisitions, regulatory changes, financial health, company growth potential, product launches, and issues within the supply chain. Each factor can result in one of two potential outcomes. For example, a 'Positive Outlook' on regulatory changes can open up new business opportunities, whereas 'Unknown or Uncertain' could signify regulatory uncertainties that lead to financial challenges.

- **Historical Financial Metrics.** Important metrics include historical earnings per share (EPS), revenue trends, and past stock price movements. Each factor can result in three outcomes: 'Bullish', where metrics like earnings per share, revenue, and stock prices consistently rise,

indicating strong financial health; 'Stable', characterized by steady movements; 'Bearish,' marked by declining financial figures, possibly leading investors to be pessimistic about the company's future performance.

We make use of the structured output capability of `GPT-4o-2024-08-06` to extract factor profiles from earnings call transcripts. Following the framework set by Liu et al. (2024), we provide the LLM with a list of factors, their potential outcomes, and associated verbalized likelihoods. For each factor, the analysis involves two steps: first, the LLM creates a concise summary specific to that factor from the transcript; second, it assigns a verbalized likelihood to each possible outcome, ranging from "very unlikely" to "very likely." Specifically, the likelihoods of outcomes, such as EPS, revenue trends, and historical stock prices, are derived from the company's historical financial data. An example of the factor profile is shown in Figure 1, and the prompts used are detailed in the Appendix.

To convert these categorical likelihoods into probabilities, we employ the following normalization process: let $P_{i,j}$ denote the likelihood associated with the $j$-th outcome for the $i$-th factor. Here, verbalized likelihoods are converted to numerical values using the mapping {very unlikely=1, unlikely=2, somewhat unlikely=3, somewhat likely=4, likely=5, very likely=6}. Then, the probability $P(O_{ij}|X)$ is calculated as $P(O_{ij}|X) = \frac{P_{i,j}}{\sum_k P_{i,k}}$, ensuring the sum of outcomes for each factor equals 1. Alternative techniques, such as instructing the LLM to "distribute 10 points among the outcomes", have been explored (Yang et al., 2024), our initial evaluation reveals that using verbalized likelihoods followed by normalization improves prediction accuracy compared to these direct probability distribution methods.

## 2.2 Analyzing Key Factors Using the Bradley-Terry Model

The Bradley-Terry model is a probabilistic framework used for estimating the relative strengths of items based on pairwise comparisons, and the outcome of each comparison indicates which of the two items is 'better' in a specific context (Bradley & Terry, 1952). This model has been widely used for ranking purposes in sports tournaments, LLM preference studies, and other domains where pairwise comparison data is available (Hu et al., 2023; Zhu et al., 2024). In this model, we estimate parameters that represent the strength of each factor. These parameters are generally presented on a logistic scale, where the probability that factor A is considered more significant than factor B is modeled as:

$$P(A > B) = \frac{e^{\beta_A}}{e^{\beta_A} + e^{\beta_B}} \tag{1}$$

Here, $\beta_A$ and $\beta_B$ represent the strengths of factors A and B, respectively. The estimated parameters are often exponentiated, so that $p_i = e^{\beta_i}$ measures the relative strength of each factor. A higher value indicates a stronger influence. In determining which factors to prioritize in a post-earnings analysis, those with higher Bradley-Terry scores are considered more crucial.

Consider a comparative analysis of two earnings call transcripts, A and B, transcript A is more likely to lead to favorable stock movements than transcript B ($A \succ B$). We obtain such pairwise comparisons based on target labels; with 'strong-buy' ranked higher than 'hold', 'sell', and 'strong-sell'; 'buy' outranking 'sell' and 'strong-sell'; and 'hold' surpassing 'strong-sell'. The comparison of A and B will involve creating a set of factor-outcome pairwise comparisons, where each outcome in transcript A is preferable to that in transcript B: $O_{\cdot,\cdot}^{(A)} \succ O_{\cdot,\cdot}^{(B)}$, suggesting that the factors associated with transcript A outperform those in transcript B.

We further consider the weight-adjusted effect of comparisons between factors. Our method compares the influence of factors from transcripts A and B by calculating an 'expected occurrence', which is determined by multiplying the likelihood of these factors appearing in both transcripts, $P(O_{ij}|X^{(A)}) \times P(O_{ij}|X^{(B)})$. This approach provides a probability-based comparison, offering a more detailed evaluation than simple counting methods. These expected occurrences then feed into a Bradley-Terry model matrix $W$. The model helps to estimate the relative importance of each factor by assigning a coefficient $p_x$ to each outcome $O_{ij}$, indicating its influence on stock investment decisions. We refine these estimates using an EM-like algorithm, which iteratively adjusts and normalizes $p_x$ to best fit the observed data.

$$p_x' = W_x \left( \sum_{y \neq x} \frac{w_{xy} + w_{yx}}{p_x + p_y} \right)^{-1} \qquad p_x = \frac{p_x'}{\sum_{y=1}^M p_y'} \tag{2}$$

## 3 BAYESIAN DECISION-MAKING

In Bayesian decision-making, utility functions play a crucial role in navigating uncertainty (Halawi et al., 2024; Lin et al., 2024; Ye et al., 2024). A Bayesian framework updates beliefs about possible outcomes. Decisions are then made by evaluating the expected utility for each possible action, which involves calculating the utility across the updated beliefs. This method ensures that choices are made to maximize expected utility, so decisions are aligned with the decision-maker's preferences and risk tolerance.

Concretely, to compute $P(O_{ij}|X)$, we construct a probabilistic factor profile from a given earnings call transcript, where $O_{ij}$ represents the $j$-th outcome of the $i$-th factor. The likelihood $P(Y|O_{ij})$, which estimates how the $j$-th outcome influences stock investment decisions, is calculated using the Bradley-Terry model. This model provides a framework for quantifying the impact each factor outcome has on the decision-making process. Using these probabilities, the Bayesian decision-making formula integrates over all factors and their potential outcomes to determine the optimal action. The overall decision is derived by:

$$\hat{Y} = \arg\max_Y \sum_i \sum_j P(Y|O_{ij})P(O_{ij}|X) \tag{3}$$

The parameters calculated by the Bradley-Terry model for $P(Y|O_{ij})$ help us determine how each factor influences stock movements. During our testing phase, transcripts are assigned to one of five decision categories based on their computed scores. For example, if the ground truth indicates there are $k$ 'strong buy' recommendations, the top $k$ scoring transcripts are classified correspondingly as 'strong buy'. This approach uses probabilistic factor profiles in conjunction with Bradley-Terry modeling to identify influential factors, providing a transparent method for understanding decision-driving elements. Moving forward, we extend beyond individual factors by examining analogous cases that directly influence decisions.

## 4 ANALOGICAL REASONING

Analogical reasoning, which involves drawing parallels between similar situations (Webb et al., 2023; Ozturkler et al., 2023; Yuan et al., 2024; Sourati et al., 2024; Yasunaga et al., 2024), is an effective method for decision-making. This approach is particularly useful when analyzing how stocks react to earnings announcements by referencing past, similar events. For example, in the tech sector, stocks often show high volatility after earnings calls that introduce significant technological updates, even if the revenue and EPS meet expectations. If a tech company is rumored to discuss a new technology trend in its upcoming earnings announcement, using this method, we can infer that this company's stock might also experience increased volatility. Investors might use this analysis to make investment decisions or hedge against potential volatility.

Accurately identifying analogous examples from earnings call transcripts is crucial. We propose a method that utilizes probabilistic factor profiles, denoted as $P(O_{ij}|X)$, where $O_{ij}$ represents the $j$-th outcome of the $i$-th factor. To measure the similarity between profiles, we calculate the Kullback-Leibler (KL) divergence, which quantifies the information loss when one probability distribution approximates another. The KL divergence is computed as follows:

$$D_{KL}(P||Q) = \sum_{i=1}^n \sum_{j=1}^m P(O_{ij}|X) \log \frac{P(O_{ij}|X)}{Q(O_{ij}|X_c)} \tag{4}$$

Here, $P$ represents the factor profile for the target transcript, and $Q$ denotes the profile for a comparative transcript $X_c$ from our training set. Transcripts with lower KL divergence values are considered more analogous, and therefore more likely to influence investor decisions similarly.

During testing, we identify the Top-K profiles that show the least divergence from a test instance's profile and present these as analogical examples for the LLM to consider when reasoning about stock movements. The LLM is asked to select the most analogous example from the Top-K and carefully evaluates the current test instance to make its prediction. This approach ensures that the alignment between profiles is contextually appropriate, thereby drawing meaningful comparisons across different transcripts. By focusing on factor profiles rather than full transcripts or their summaries, we emphasize key market-moving information, avoiding unnecessary details. For example, Google and Broadcom

| System | Recall | Prec. | $F_1$ | Accu. | Label | Recall | Prec. | $F_1$ |
|---|---|---|---|---|---|---|---|---|
| LLM+CoT+Trans | 21.56 | 33.66 | 13.52 | 19.59 | Strong Sell | 7.32 | 37.50 | 12.24 |
| LLM+CoT+Summ | 22.77 | 16.17 | 14.12 | 20.61 | Sell | 5.56 | 9.09 | 6.90 |
| LLM+CoT+Factors | 24.38 | 28.58 | 17.26 | 22.32 | Hold | 29.84 | 28.24 | 29.02 |
| DeLLMa (Liu et al., 2024) | 38.30 | 23.14 | 16.68 | 22.35 | Buy | 44.83 | 18.93 | 26.62 |
| **DEFINE (Ours)** | 26.15 | 27.67 | **23.73** | **29.64** | Strong Buy | 43.22 | 44.56 | **43.88** |

Table 3: (*Left*) We show the accuracy and macro-averaged F-scores for various systems. Our system, DEFINE, which combines factor profiles with analogical reasoning, achieves the best performance. (*Right*) DEFINE's performance across five categories: Strong Sell, Sell, Hold, Buy, and Strong Buy.

could have analogous profiles even though their discussions in earnings calls might vary widely. Using factor profiles as analogous examples also requires significantly fewer tokens within the context window than full transcripts would.

## 5 DATA COLLECTION

Our dataset contains 11,950 earnings call transcripts from S&P 500 and NASDAQ 500 companies, gathered from the Motley Fool over the period of 2017–2024. The Motley Fool is a well-regarded financial service website that regularly publishes earnings call transcripts from U.S. companies. We make sure to follow their terms of use carefully during data collection. We do not use audio recordings or analyze acoustic or prosodic features. Each transcript is formatted as a JSON object, including the company's stock ticker, the date of the earnings announcement, participant names and their affiliations, executive prepared remarks, and a series of question-answer pairs from the Q&A session. Table 2 presents the statistics

**Data Statistics**

| | |
|---|---|
| Num. of Transcripts | 11,950 |
| Num. of Companies | 869 |
| Avg. #Tokens per Transcript | 10,187 |
| Avg. #QA Pairs per Transcript | 10 |
| Avg. #Trans per Company | 14 |
| Avg. #Speakers per Transcript | 12 |
| Year Range | 2017–2024 |

Table 2: Our dataset includes 11,950 earnings call transcripts from 800+ companies.

of our dataset. Each transcript averages 10,187 tokens and 133 sentences. They are sourced from 869 companies, each contributing an average of 14 transcripts. We obtain company stock prices from Yahoo Finance via the `yfinance` package and financial metrics such as revenue and earnings per share (EPS) from Alpha Advantage. Our dataset spans from 2017 to 2024. It enhances previous studies which examined earnings call transcripts from 2002–2010 (Li et al., 2020); these earlier transcripts may already be used in LLM pretraining. To avoid data contamination, we established a new test set consisting of the most recent 587 transcripts from 2024, which are beyond the pretraining cut-off date for LLMs.

We seek to make stock investment decisions by analyzing earnings call transcripts and focusing on performance over the 30-day period. We establish the ground truth decision on the 30th day following each earnings announcement (Sonkiya et al., 2021): a stock drop exceeding 5% corresponds to a 'strong sell' decision, a decrease between 2% and 5% leads to a 'sell', fluctuation within -2% to +2% is labeled 'hold', an increase between 2% and 5% is labeled a 'buy', and an increase above 5% is a 'strong buy'. In our test set, the distribution of these labels is as follows: 'strong buy' at 34%, 'buy' at 15%, 'hold' at 21%, 'sell' at 9%, and 'strong sell' at 21%. This distribution is generally balanced, reflecting a slightly bullish market trend in 2024.

## 6 EXPERIMENTS

In this section, we evaluate the decision-making performance of various systems, analyze the key factors that influence stock movement predictions, and conduct an analysis of analogical reasoning.

### 6.1 DECISION-MAKING WITH DEFINE

We test our system, DEFINE, against different decision-making strategies: (a) ***LLM+CoT+Trans***: We feed the entire earnings call transcript to the LLM and then use the chain-of-thought to assign a

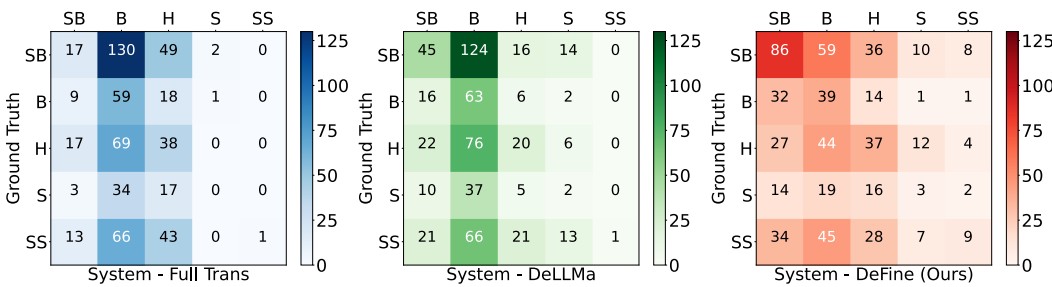

Figure 2: A comparison of confusion matrices from the LLM+CoT+Trans, DeLLMa, and DEFINE methods. While LLM+CoT+Trans and DeLLMa lean towards 'Buy (B),' DeFine offers more balanced outcomes across all decision categories, showing notable improvement in 'Strong Buy (SB),' 'Buy (B),' 'Hold (H),' and 'Sell (S)' decisions.

label, with both labels and their interpretations also provided to the LLM. (b) **LLM+CoT+Summ** and **LLM+CoT+Factors**: These approaches use a summarize-then-predict strategy. LLM+CoT+Summ simplifies the transcript into a textual summary, while LLM+CoT+Factors condenses it into a factor profile. It is considered a structured summary, unlike the textual summary produced by direct LLM prompting. Details on the prompts used for these methods can be found in the Appendix.

Our system, DEFINE, utilizes analogical reasoning by analyzing five analogous cases identified using KL-divergence as the distance metric. It examines these cases alongside the current factor profile to predict an appropriate label. In contrast, DeLLMa uses a decision theory approach and has shown strong performance in agriculture planning and finance (Liu et al., 2024). For this approach, we pair each factor profile with possible labels and choose the top-ranked outcome as the final decision.

In Table 3 (left), we present the accuracy and macro-averaged F-scores for various systems, all using `GPT-4o-2024-08-06`. Our new system, DEFINE, which combines factor profiles with analogical reasoning, achieves the best performance. It surpasses the strong baseline system, DeLLMa, which involves ranking state-action pairs based on their preference levels as determined by the LLM. We find that LLMs generally make more accurate decisions when working with summaries rather than full transcripts; those transcripts typically contain around 10k tokens. This finding underscores the complexity of extracting and weighing key factors from lengthy transcripts, a task that remains challenging for most LLMs. In contrast, our factor profile method proves advantageous as it provides a balanced view of both macroeconomic factors and company-specific details, which are essential for rational decision-making.

We further analyze DEFINE's performance across five categories: Strong Sell, Sell, Hold, Buy, and Strong Buy. Results are shown in Table 3 (right). DEFINE performs best at 'Strong Buy' recommendations and faces challenges with 'Strong Sell' categories. This may be due to its reliance on earnings call transcripts, which often contain optimistic remarks from executives aimed at reassuring investors, potentially skewing predictions away from 'Strong Sell.' Figure 2 includes a comparison of confusion matrices from the LLM+CoT+Trans, DeLLMa, and DEFINE methods. While LLM+CoT+Trans and DeLLMa predominantly lean towards 'Buy,' DeFine offers more balanced outcomes across all decision categories, showing notable improvement in 'Strong Buy,' 'Buy,' 'Hold,' and 'Sell' decisions.

## 6.2 INFLUENTIAL FACTORS

We develop three variations of our DEFINE-BT approach, each using the Bradley-Terry model for pairwise comparisons in different contexts: DEFINE-BT-Same Sector compares companies within the same sector, DEFINE-BT-Cross Sectors examines companies across different sectors, and DEFINE-BT-Same Company analyzes a company's current earnings call transcript against its historical ones. To ensure fairness, we maintain the same number of pairwise comparisons across all three settings, downsampling where necessary. According to the F-scores presented in Table 4, all DEFINE-BT variants outperform both the random baseline, which assigns investment decisions randomly from five possible labels, and DeLLMa on the test set.

Among the three variants, DEFINE-BT-Cross Sector achieves the highest scores in both F-Score and Accuracy. This indicates that considering pairwise comparisons between earnings announcements

| Factor (Outcome) | Salience |
|---|---|
| - Regulatory changes and legal issues happened (positive outlook) | 0.0364 |
| - Natural disasters and other black swan events (major impact) | 0.0360 |
| - Political events and government policies (major upheaval) | 0.0349 |
| - Geopolitical issues (escalation to conflict) | 0.0345 |
| - Supply chain (positive outlook) | 0.0322 |
| - Tech innovation (positive outlook) | 0.0317 |
| - Historical stock price change (bullish) | 0.0316 |
| - Historical EPS (bullish) | 0.0315 |
| - Financial health (positive outlook) | 0.0311 |

Table 5: Influential factors and outcomes that drive bullish investment decisions in the ***Consumer Defensive*** sector, such as food and beverage, household products, and grocery stores.

| Factor (Outcome) | Salience |
|---|---|
| - Economic health (unknown or uncertain) | 0.0362 |
| - Market sentiment and investor psychology (unknown or uncertain) | 0.0350 |
| - Company growth (unknown or uncertain) | 0.0338 |
| - Supply chain (unknown or uncertain) | 0.0326 |
| - Geopolitical issues (escalation to conflict) | 0.0322 |
| - Historical revenue (decline) | 0.0319 |
| - Historical stock price change (bullish) | 0.0318 |
| - Tech innovation (unknown or uncertain) | 0.0315 |
| - Natural disasters and other black swan events (major impact) | 0.0315 |
| - Political events and government policies (major upheaval) | 0.0313 |

Table 6: Factors and outcomes that drive bullish investment decisions in the ***Technology*** sector, including industry leaders such as Apple, Microsoft, Amazon, Google, and Meta.

from a diverse range of companies can enhance predictions of stock movements. Table 7 illustrates the performance of DEFINE-BT-Cross Sector, which was trained on one sector and tested on another. For this analysis, 100 earnings call transcripts were selected from each of the 11 financial sectors: Technology, Healthcare, Financial Services, Consumer Defensive, Energy, Industrials, Utilities, Basic Materials, Real Estate, Consumer Cyclical, and Communication Services.

Tables 5 and 6 highlight influential factors impacting investment decisions in the Consumer Defensive and Technology sectors, as identified by the Bradley-Terry model. In Consumer Defensive, which includes industries like food and beverage, household products, and grocery stores, significant drivers are natural disasters and black swan events, political events and government policies, and geopolitical issues. These challenging macroeconomic circumstances often lead to buy-in decisions from investors. In contrast, the Technology sector, with industry leaders such as Apple, Microsoft, Amazon, Google, Meta, and Nvidia, shows that decisions to invest often hinge on unclear or uncertain factors. Technology stocks have seen considerable growth from 2017–2024. This pattern suggests that investment models may favor purchases in these companies despite encountering negative issues in earnings announcements.

|  | $F_1$ | Accu. |
|---|---|---|
| Random Baseline | 18.00 | 19.11 |
| DeLLMa ((Liu et al., 2024)) | 16.68 | 22.53 |
| DEFINE-BT-Same Sector | 20.11 | 22.15 |
| DEFINE-BT-Same Company | 20.42 | 23.68 |
| DEFINE-BT-Cross Sectors | **24.45** | **27.43** |

Table 4: Among the three variants, DEFINE-BT-Cross Sector achieves the highest scores, suggesting that considering pairwise comparisons from a diverse range of companies can enhance the predictions of stock movements.

In Figure 4, we analyze the probability of positive and negative factor outcomes, represented as a continuous random variable, and plot its probability density function (PDF) for various investment decisions. Highlighted sections illustrate where the gaps between strong buy (red) and strong sell (blue) decisions are most pronounced. Our analysis indicates that buy decisions often occur when the probability of positive outcomes is relatively low (about 0.2-0.3) and the likelihood of negative outcomes is moderate to high (ranging from 0.3 to 0.65), but not overly negative. Conversely, sell decisions tend to occur when negative outcome probabilities are minimal (about 0.1-0.2). These observations suggest that rational investment decisions can sometimes appear counterintuitive: essentially, selling high and buying low. We find that a thorough analysis of various factors is advantageous. Our approach incorporates not just the known issues but also the uncertain or hidden factors, thereby enhancing the decision-making process.

## 6.3 INSIGHTS INTO ANALOGICAL REASONING

Analogical reasoning utilizes a select number of analogous examples, denoted as $K$, to inform decision-making in LLMs. In Figure 3, we adjust $K$ from 3 to 9 and observe its impact on the F-Score. In these experiments, we use the majority vote from the $K$ examples as the final prediction. We find that $K = 4$ achieves the highest performance, potentially due to some tie-induced randomness compared to odd numbers. Typically, odd numbers for $K$ are preferred for majority voting to avoid ties, with $K = 3, 5, 7$ showing similar effectiveness. For our system, DEFINE, we have opted

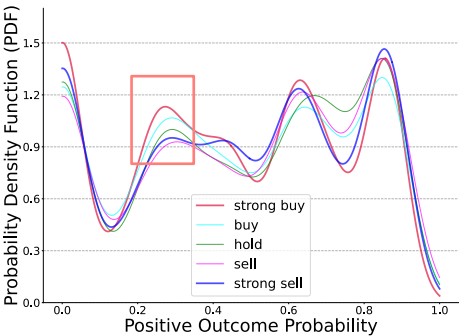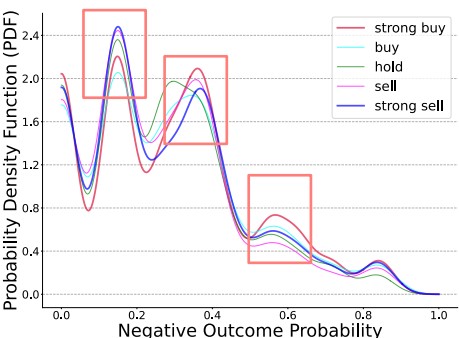

Figure 4: We analyze and plot the probability density function (PDF) of positive and negative factor outcomes for different investment decisions. Highlighted sections illustrate where the gaps between strong buy (red) and strong sell (blue) decisions are most pronounced.

for $K = 5$ to strike a balance between providing enough analogous examples and maintaining a manageable context length for the LLM.

Moreover, we examine how the most analogous examples influence DEFINE's predictions. Our study finds that in 69% of cases, the LLM's predictions match the labels from the most analogous examples. In the other 31% of cases, the LLM chooses to make its own predictions. E.g., when the analogous example is labeled "Strong Buy," DeFine concurs with "Strong Buy" in 63% of cases. It opts for "Buy" in 26% and "Hold" in 11% of the cases. Conversely, when the example is "Strong Sell," DEFINE agrees with "Strong Sell" 50% of the time, chooses "Sell" in 25% of cases, and "Hold" in 12.5%. These results indicate that while DEFINE effectively utilizes analogous historical data to inform its predictions, it also critically evaluates the current factor profiles, demonstrating a balanced approach in its decision-making abilities.

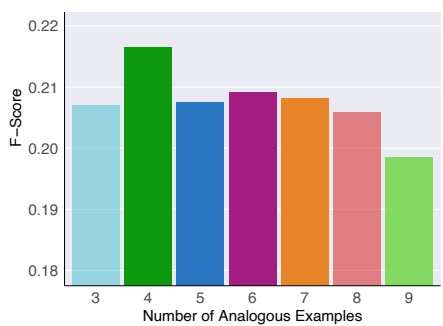

Figure 3: Analogical reasoning utilizes a number of analogous examples to inform decision-making in LLMs. We adjust $K$ from 3 to 9 and observe its impact on the F-Score.

## 7 RELATED WORK

**Analogical Reasoning.** This type of reasoning identifies connections between similar, though not identical, situations to transfer knowledge from a known context to a new one (Webb et al., 2023; Ozturkler et al., 2023; Yu et al., 2024; Yuan et al., 2024; Sourati et al., 2024; Yasunaga et al., 2024). It helps decision makers draw parallels between current situations and past experiences, effectively leveraging historical insights. Analogical reasoning plays a crucial role in various fields, e.g., doctors apply knowledge from one disease to diagnose another, and lawyers use past rulings to argue new cases (Lehman et al., 2022; Charmet et al., 2022; Cao et al., 2024a). This ability to recognize and use similarities in different situations is important for decision-making.

While zero-shot analogical reasoning is a desired capability for LLMs, recent studies show they lack the robustness and generality of human analogy-making, as evidenced by counterexamples in tasks such as letter string analogies (Hodel & West, 2024; Lewis & Mitchell, 2024). Musker et al. (2024) test both humans and LLMs on tasks that require transferring semantic structure and content between domains. Yasunaga et al. (2024) introduce analogical prompting, where LLMs self-generate relevant examples using prompts such as "*# Recall relevant problems and solutions:*" before solving the original problem; Qin et al. (2024) find that the accuracy of self-generated examples is key to eliciting such capability. Unlike previous research, our study employs probabilistic factor profiles to model analogical reasoning, grounding our approach in solid mathematical principles.

**LLM Decision-Making under Uncertainty.** The use of LLMs in decision-making has surged due to their remarkable ability to reason over complex scenarios (Halawi et al., 2024; Lin et al., 2024; Ye

| | Tech | FS | Health | CC | Ind | CS | CD | Energy | RE | BM | Util |
|---|---|---|---|---|---|---|---|---|---|---|---|
| Technology (Tech) | 15.40 | 17.99 | 17.39 | 12.10 | 13.15 | 18.19 | 25.85 | 27.09 | 13.82 | 22.86 | 26.67 |
| Financial Services (FS) | 15.96 | 17.96 | 26.84 | 7.99 | 10.21 | 26.22 | 15.45 | 4.80 | 13.37 | 21.37 | 0.00 |
| Healthcare (Health) | 16.73 | 19.80 | 17.89 | 21.85 | 28.46 | 10.86 | 20.23 | 18.73 | 3.64 | 43.45 | 73.33 |
| Consumer Cyclical (CC) | 18.14 | 11.02 | 19.38 | 19.39 | 15.86 | 9.49 | 17.70 | 17.40 | 12.19 | 22.22 | 36.67 |
| Industrials (Ind) | 17.02 | 11.14 | 14.37 | 11.24 | 18.81 | 15.93 | 19.48 | 25.11 | 3.20 | 24.44 | 0.00 |
| Communication Services (CS) | 18.61 | 14.68 | 18.87 | 14.03 | 19.47 | 33.70 | 10.71 | 16.99 | 11.87 | 10.26 | 13.33 |
| Consumer Defensive (CD) | 24.91 | 21.71 | 19.15 | 19.89 | 21.38 | 2.67 | 23.09 | 12.50 | 9.72 | 29.52 | 50.00 |
| Energy | 19.49 | 16.50 | 23.62 | 14.25 | 19.03 | 8.90 | 19.03 | 8.98 | 12.10 | 28.79 | 0.00 |
| Real Estate (RE) | 22.86 | 15.74 | 16.76 | 14.08 | 12.34 | 4.00 | 15.61 | 11.28 | 12.34 | 43.18 | 0.00 |
| Basic Materials (BM) | 20.67 | 13.69 | 15.26 | 18.18 | 29.64 | 9.52 | 21.19 | 17.19 | 17.10 | 16.67 | 37.50 |
| Utilities (Util) | 17.82 | 27.75 | 23.15 | 26.25 | 12.61 | 25.49 | 20.63 | 5.70 | 18.40 | 14.29 | 53.33 |

Table 7: The performance of DEFINE-BT was evaluated by training it on one financial sector and testing it on another using 100 earnings call transcripts from each of the 11 sectors.

et al., 2024; Band et al., 2024). However, the challenge of balancing a multitude of often conflicting factors in decision making remains understudied. For example, Falck et al. (2024) investigate whether adding more data points in in-context learning reduces uncertainty, as typically expected in Bayesian learning, and find evidence against this theory. The DeLLMa framework (Liu et al., 2024) incorporates uncertainty into LLM decision-making using Bayesian networks and has been tested on tasks such as agriculture planning and finance. Feng et al. (2024) employ LLM entailment to map factors to context and utilize trained Bayesian models for probability estimation. Our work builds on these initiatives by integrating analogical reasoning with factor profiles to enhance the accuracy and transparency of LLM decision-making.

**Financial Forecasting.** Recent advancements in LLMs have revolutionized traditional financial tasks (Keith & Stent, 2019; Sawhney et al., 2020; 2021; Chuang & Yang, 2022; Ang & Lim, 2022; Sang & Bao, 2022; Medya et al., 2022; Wang et al., 2023; Koa et al., 2024; Srivastava et al., 2024). Notably, Chen et al. (2022) introduce FinQA, a dataset constructed from financial statements for assessing LLMs' multi-step numerical reasoning. Moreover, TAT-QA (Zhu et al., 2021) tackles QA over tabular and textual data; FiNER (Loukas et al., 2022) focuses on numerical entity recognition; DocFinQA (Reddy et al., 2024) is a dataset designed for long-document financial QA; RiskLabs (Cao et al., 2024b) employs LLMs for financial risk assessments. Nie et al. (2024) provide a comprehensive survey on the use of LLMs across various financial domains. Our study focuses on analyzing earnings transcripts to understand how LLMs handle the ambiguities inherent in spoken language, thus providing insight into their decision-making under uncertainty. The research findings have broader applications including medical consultations, negotiations, and political debates.

# 8 CONCLUSION

We propose DEFINE, a new framework for decision-making in complex scenarios, such as those encountered in corporate earnings calls. By combining probabilistic factor profiles with analogical reasoning, this framework not only captures the uncertainties embedded in earnings call transcripts but also allows the LLM to apply previous insights to new challenges more efficiently. Our approach surpasses strong baseline models and enhances the practical utility of LLMs by identifying analogous examples. The DEFINE framework offers a promising avenue for navigating complex data and supporting decision-making processes.

# 9 LIMITATIONS

The effectiveness of the DEFINE framework, as presented in this paper, is predominantly based on controlled experimental conditions. While the framework has been designed to enhance decision-making capabilities through the use of probabilistic factor profiles and analogical reasoning, actual outcomes may vary when applied in real-world scenarios. Users should be aware that the framework's performance can be influenced by various external factors including data quality, context-specific nuances, and the dynamic nature of real-world environments. We encourage users to consider these variables when implementing and adapting the DEFINE approach to ensure its optimal application and to mitigate potential discrepancies between expected and actual results.

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

# A APPENDIX

---

**Constructing a Factor Profile from an Earnings Call Transcript**

**System Message**

You are a financial analyst specializing in earnings call transcripts. You will receive the complete transcript of an earnings call, which includes both the prepared remarks and the Q&A session. Your job is to identify the key factors from the transcript and assign probabilities to the potential outcomes of these factors.

**User Message**

Your task is to conduct a comprehensive analysis of the earnings call transcript below. Be sure to accurately capture the important factors and estimate the likelihood of each factor resulting in specific outcomes.

Earnings Call Transcript for Company {Company}
# Prepared Remarks
Speaker: {Speech...}
...

# Questions and Answers
Analyst: {Speech...}
...

Please analyze the above earnings call transcript, focusing on the following key factors:
{Enumerate factors, descriptions, and outcomes}

1. Economic health: Economic health refers to the overall stability and performance of the economy, reflected in factors like growth, employment, inflation, and market confidence. Outcomes: {positive-outlook, unknown-or-uncertain}

2. Market sentiment and investor psychology: Market sentiment reflects the overall mood or attitude of investors toward a particular market, influenced by news, economic data, and global events. Investor psychology refers to the emotions and cognitive biases that drive decisions, often leading to behaviors like fear-driven selling or greed-fueled buying. Outcomes: {optimistic, unknown-or-uncertain}
...

Please take the time to thoroughly understand the transcript. For each key factor, provide a detailed summary based on the given transcript. Then, review all associated outcomes and assess the likelihood of each outcome. The likelihood should be strictly selected from the following options: {very likely, likely, somewhat likely, somewhat unlikely, unlikely, very unlikely}. Format your response in JSON.

# Example Output:
{JSON output example}

# Your Output:

Figure 5: **Constructing a Factor Profile from an Earnings Call Transcript**

---

---

### Using Analogical Reasoning to Make Investment Decisions

**System Message**

You're a financial analyst who specializes in giving investors buy or sell recommendations by thoroughly analyzing earnings call transcripts.

**User Message**

Here are several example company profiles. Each profile highlights key factors from an earnings call transcript and probabilities for potential outcomes based on those factors. Each profile represents a specific company and is based on its historical earnings call data. Your job is to pick the most analogous example and use its strategy to solve the initial problem.

Example Company Profile 1:
{Factor Profile 1}
Analyst recommendation: {Action 1}
Example Company Profile 2:
{Factor Profile 2}
Analyst recommendation: {Action 2}
Example Company Profile 3:
{Factor Profile 3}
Analyst recommendation: {Action 3}
Example Company Profile 4:
{Factor Profile 4}
Analyst recommendation: {Action 4}
Example Company Profile 5:
{Factor Profile 5}
Analyst recommendation: {Action 5}

**Initial Problem**

Based on your analysis of the earnings call for {Company Name} held on {Announcement Date}, decide on the most likely analyst recommendation for the next 30 days from these options:

 - Action 1: strong buy: The stock price will increase by more than 5%
 - Action 2: buy: The stock price will increase by 2% to 5%
 - Action 3: hold: The stock price is expected to remain stable, fluctuating between -2% to 2%
 - Action 4: sell: The stock price will decrease by 2% to 5%
 - Action 5: strong sell: The stock price will decrease by more than 5%

Below is the company profile summarized from {Company Name}'s earnings call on {Announcement Date} and the historical price trend probabilities judged by an analyst:

{Factor Profile Constructed Using an Earnings Call Transcript}

**Solve the Initial Problem**

Please respond with the analyst recommendation for this stock in JSON format, including these keys: ('idx', 'recommendation', 'justification'). 'idx' is the index of the most analogous example profile, and 'recommendation' should be one of the actions mentioned above for 30 days of trading, and 'justification' should clearly explain your recommendation using the strategy you learned from the selected example company profile.

Figure 6: **Using Analogical Reasoning to Make Investment Decisions**

---

### A Prompt that Uses Chain-of-Thought to Make Investment Decisions

**System Message**

You're a financial analyst spcializing in giving investors buy or sell recommendations by thoroughly analyzing earnings call transcripts.

**User Message**

Based on your analysis of the earnings call for {Company Name} held on {Announcement Date}, decide on the most likely analyst recommendation for the next 30 days from these options:

- Action 1: strong buy: The stock price will increase by more than 5%
- Action 2: buy: The stock price will increase by 2% to 5%
- Action 3: hold: The stock price is expected to remain stable, fluctuating between -2% to 2%
- Action 4: sell: The stock price will decrease by 2% to 5%
- Action 5: strong sell: The stock price will decrease by more than 5%

Below is the {Factor Profile, Transcript or Summary} from {Company Name}'s earnings call on {Announcement Date}:}

{Factor Profile, Transcripts or Summary}

Please think step by step and respond with the analyst recommendation for this stock in JSON format, including these keys: ('thoughts', 'recommendation', 'justification'). 'Thoughts' should be your detailed reasoning steps, 'recommendation' should be one of the actions mentioned above for 30 days trading, 'Justification' should clearly explain your recommendation using the strategy you learned from the selected example company profile.

Figure 7: **A Prompt that Uses Chain-of-Thought to Make Investment Decisions**

---

### A Prompt to Analyze Trends Based on Historical Financial Metrics

**System Message**

You are a financial analyst specializing in historical data analysis, including stock prices, earnings per share (EPS), and revenue. Your goal is to assess the likelihood of different market trends based on past data.

**User Message**

The potential outcomes to consider are: {bullish, stable, and bearish}. For each outcome, please assign a likelihood level from the following options: {very likely, likely, somewhat likely, somewhat unlikely, unlikely, very unlikely}.

Below, you will be provided with a historical data table: {Data Name}:{Description}
{Historical Data Table}

```
  Date        Close Price
  2023-07-31  195.22
  2023-08-01  195.46
{... until the date of the earnings announcement.}
```

Please analyze this historical data and provide the likelihood of each outcome in JSON format.

# Example Output:
{"historical EPS":{"bullish": very likely, "stable":somewhat likely, "bearish": unlikely}}

# Your Output:

Figure 8: **A Prompt to Analyze Trends Based on Historical Financial Metrics**

