# OpenReview forum: "DeFine: Enhancing LLM Decision-Making with Factor Profiles and Analogical Reasoning"
_ICLR.cc/2025/Conference — Submitted to ICLR 2025_

### Official Review · Reviewer_5EPe · 2024-10-29

**Soundness:** 3
**Presentation:** 4
**Contribution:** 3
**Rating:** 6
**Confidence:** 3

**Summary:**

This paper proposes a novel framework,DEFINE, combining Probabilistic Factor Profile and Analogical reasoning in LLMs to support investment decisions. By using LLMs to extract factor profiles in transcripts, the model captures key information and their outputs in relevant domains. These factor profiles are utilized as analogous examples when the model is tested in unseen scenarios. By providing investment decisions like buying or selling stocks, the authors found that the model outperformed traditional models like DeLLMA and CoT-prompted LLMs. The downstream evaluation also shows that the model performs well in cross-sector domains with the application of analogical reasoning. The authors also investigate the most influential factors in the Consumer Defensive and technology domain, revealing the key information that supports the model's investment decisions. This study sheds light on the application in financial and commercial situations.

**Strengths:**

- The authors propose a framework, combining Probabilistic Factor Profiles with Analogical reasoning,  endowing the model with a strong generalizability in unseen scenarios and even cross-domain sectors.

- The authors compared five models (including DEFINE) with their performance in making 'correct' investment decisions, which their model outperforms other models in the accuracy metrics. Downstream evaluations are also applied in cross-domain sectors, demonstrating superior performance than random chance level, which makes the result more robust and generalizable.

- The authors also investigate the mechanisms and details of how this model works. By using the Bradley-Terry model, the authors figure out the most important factors from the profiles. The authors also explore the number of optimal analogous examples to provide in the test phase. Finally, the authors also find out the associations between the outcomes in the profile and investment decisions. Overall, the mechanistic analysis shows a step-by-step flow of how the model works.

- The paper's writing and visualization is satisfying.

**Weaknesses:**

- One important contribution of the paper is the model adopts a framework with Probabilistic Factor Profiles with Analogical Reasoning. However, in the model comparison, three baseline LLMs are prompted with CoT with different instructions. If I do not understand incorrectly, these models do not contain any training or in-context learning to do the task. While factor profiles may be a good way to do so, I am wondering how well the LLM with purely CoT, with some examples in the context (just like transcripts and their optimal decisions) will perform. This aims to figure out the role of Analogical reasoning in the DEFINE model, which is tested in a more general sense. It is possible that LLM can do analogical reasoning themselves, without showing them explicitly.

- Currently, the model uses analogous examples based on KL divergence between the current scenario and stored profiles. This is a good intuition to refer but it could strengthen the robustness if more ways are into consideration and compared. For example, there can be another LLM to 'help'  to pick up five examples from the profiles that it thinks referable. Or train a model based on the downstream accuracy and then test the whole pipeline on OOD datasets.

The authors do not need to do all the proposed analysis but it would be better to clarify the exact role of the two key components in the new model.

**Questions:**

- One observation for this paper is though the model outperforms other model candidates, the increasing part mainly emerges in the action 'strong buy' for 'sell' actions, the model still fails to figure them out. What could be possible reasons for this phenomenon? Training Data distribution? Or model pipeline?

- Another similar question is why some domains are better in cross-domain but a few of them are not.

These questions are just proposed for further public discussion and do not necessarily mean any supplementary analysis.

---

> ### Author Response · Authors · 2024-11-23
> **Appreciate the Insights from Reviewer 5EPe**
>
> __W1__
>
> Thanks for this thoughtful question! We completely agree that exploring how LLMs perform with a few in-context examples (without analogical reasoning) would provide valuable insights into the role of analogical reasoning in DeFine.
>
> There are some practical challenges to running such experiments. Each earnings call transcript in our dataset contains about 10k tokens. Incorporating multiple full transcripts with their decisions as context would be computationally prohibitive. That said, we'd love to explore this direction in future work, possibly starting with a smaller-scale study to shed light into how much analogical reasoning is 'implicit' in LLMs!
>
> __W2__
>
> Awesome suggestions! We really appreciate your thoughtful input. Using KL divergence to identify analogous examples has worked well in our experiments, and you're absolutely right that exploring alternative approaches could make the model even more robust.
>
> __Q1, Q2__
>
> One key factor is likely the distribution of the training data. As we mentioned in the paper, the stock market from 2017 to 2024 has been slightly bullish overall, with more instances favoring positive outcomes such as 'buy' and 'strong buy.' This trend might have influenced the model to prioritize optimistic predictions, even when there are negative signals in earnings announcements. Thank you for pointing this out!
>
> We hope our responses have addressed your questions and clarified the strengths of our approach. If this meets your expectations, we'd be so grateful if you could consider raising your overall rating. Your support would mean a lot to us!

---

> > ### Comment · Reviewer_5EPe · 2024-11-26
> >
> > Thanks for the clarifications and explanations (for Q1 and Q2) from the authors.
> >
> > For the w1 response, I mean to use the constructed profiles (whether by experts or by LLMs themselves) as examples to do in-context learning. Putting raw transcripts may be hard but having LLMs directly work on serveral profiles may also 'emerge' analogical reasoning in actual decision-making, which could be a useful baseline. All the suggestions here are to confirm the roles of analogical reasoning and KL divergence, which are robust and necessary to solve this problem.
> >
> > Generally, this work is good and I think it should be considered acceptance. However, given the overall comparison, the confidence in robustness could be strengthened. Cross-referring other reviewers' suggestions, there are similar concerns in other aspects of the evaluation of current methods (e.g., cost-effectiveness). Therefore, I would like to maintain my rating.

---

> > > ### Author Response · Authors · 2024-11-26
> > > **Appreciate the Insights from Reviewer 5EPe**
> > >
> > > Dear Reviewer 5EPe,
> > >
> > > Thank you so much for recommending our work for acceptance! Your points are well taken, and we're happy to strengthen the robustness of our approach and include the additional experiments you suggested to confirm the roles of analogical reasoning and KL divergence in solving LLM decision-making problems. We appreciate your support!

---

### Official Review · Reviewer_2pMM · 2024-11-02

**Soundness:** 3
**Presentation:** 2
**Contribution:** 3
**Rating:** 6
**Confidence:** 3

**Summary:**

This paper presents a novel approach to financial decision-making using LLMs. The authors propose using LLMs to generate factor summaries and predict outcome probabilities from earnings call transcripts. They also develop a method to find analogous examples from their training data to support the LLM during inference. Authors compare with 3 in-house baselines and 1 additional paper, DeLLMa. They show better performance, then perform ablations and report some insights.

**Strengths:**

I think there is value in what the authors did and I like the probabilistic grounding. The setup is rather interesting and I like the more targeted selection of in-context examples. Providing LLMs with better ways to do reasoning is a fundamental task.

I appreciate the structured approach here: authors first use an LLM to extract factors/categories from transcript calls and then they associate outcomes to those.

**Weaknesses:**

Since I'm not a finance expert, I kind of can't fully assess how applicable this approach is in practice. One of my issues is about the core contribution: are the authors essentially proposing an improved method for selecting in-context examples for the language model - “better way” that is, in this case, specific to their finance domain? If so, this, to me, feels very specific and slightly incremental.

While the setup is interesting, I see several limitations in the experimental evaluation:

* With a test set of 500 examples, I'd really like to see standard deviations to understand how reliable the results are, especially for the ablation studies.

* The baseline selection isn't well justified - for instance, is there a baseline that uses the DeFine model without the analogical reasoning component? or with a similarity search of similar transcripts/related factors instead of the analogical reasoning part? I would appreciate if the authors could defend their baselines.

* How does DeLLMa work? why is this a “strong baseline”? I would at least need to see a summary in the paper since this is the only baseline coming from an external work.

* The experimental details are insufficient for reproducibility. The authors do share prompts, but I am not sure which were the temperature, and even if more experiments had been run to account for noise - I understand the authors can probably add this to the camera ready, but still. Which prompts are the ones used for the baselines? the fact that from Figure 5 onwards there is not a backref to the text makes this harder to understand (also, names in the appendix are different from what I see in the paper?)

Minor notes:
* When discussing analogical reasoning, the authors only cite work from 2022 onwards, ignoring fundamental AI literature (e.g., Minsky's 1991 paper comes to mind - which I understand might be out of scope here, but the point still stands). While this might not be their focus, it overlooks important historical context (again, I know this is not the focus of the paper).

* I am putting this into the minor notes since it could be a “me” issue, but to me this paper was a bit too dense to read. Information density is very high, and the paper would benefit from more experiments (e.g., Outcomes are introduced in 2.1 and then defined only later). I had to go up and down the paper a couple of times to recover information. Sections 3 and 4 are very short and would benefit from some restructuring. They can probably be both put under the same main section.

* Not a finance expert, but how can I interpret that F1 score? is it good? Isn’t one of the issues with market prediction the fact that it’s hard to use history to predict the future? Also, I am not sure if I missed that here, but how would a random/simpler baseline perform here?

**Questions:**

//

---

> ### Author Response · Authors · 2024-11-23
> **Appreciating the Feedback from Reviewer 2pMM and Our Clarification**
>
> We appreciate the reviewer's thoughtful questions. To clarify, our work isn't just an incremental improvement for selecting in-context examples, it's an important step toward analogical decision-making. DeFine introduces probabilistic factor profiles paired with analogical reasoning, allowing LLMs to address uncertainty in complex scenarios. While we used financial data as a high-stakes test case, the framework is modular and adaptable across domains such as medical diagnostics, where decisions under uncertainty are critical.
>
> __W1__
>
> We totally understand the importance of showing reliability with standard deviations. Our test set examples were carefully curated to ensure no overlap with LLM pretraining data (these transcripts are from 2024, well past the pretraining cut-off of October 2023). This setup ensures a clean evaluation of generalization. For the ablation studies, we'll include standard deviations in the final version to give a clearer picture of the results.
>
> __W2__
>
> Thanks for raising this! We've actually included results for DeFine without analogical reasoning in Table 6. DeFine's strength lies in its use of probabilistic factor profiles to capture uncertainty and guide decisions, so removing analogical reasoning still demonstrates satisfying performance.
>
> As for a similarity-based baseline, we found it less effective for two big reasons. First, transcript texts can differ a lot in wording, so methods such as embeddings or keyword searches often introduce noise, they focus too much on surface-level similarities rather than key determinants for decision-making. Second, these approaches can miss the nuances of uncertainty. That nuance is captured by our probabilistic factor profiles but would be overlooked in a simple text-based similarity search.
>
> We hope this clarifies the thought process behind our baselines! Let us know if there's anything else you'd like us to expand on.
>
> __W3__
>
> Great question! DeLLMa is a recently introduced framework that incorporates probabilistic modeling for decision-making under uncertainty. It's particularly strong because it combines Bayesian inference with LLM capabilities, and it's been tested on challenging tasks such as agriculture planning and finance. We chose it as a baseline because it's one of the few external models tackling similar complex scenarios. We'll add a concise summary of DeLLMa's approach in the paper to ensure clarity.
>
> __W4__
>
> We'll release our source code alongside all the prompts used in our experiments to improve reproducibility. This will make it easier for others to build on our work. For clarity, the temperature was set to zero in all experiments, and we'll explicitly add this detail in the final version.
>
> We'll also make sure figures in the Appendix have back-references to the main text. We hope this addresses your concerns and makes our work more accessible!
>
> __Minor 1-3__
>
> * You're absolutely right that analogical reasoning has deep roots in AI, and incorporating works such as Minsky's 1991 paper would add valuable historical context. While our focus is on modern methods, we'd be happy to include references to earlier research to better situate our work within the broader AI literature. Thank you for the great suggestion!
>
> * We totally get that the paper can feel dense, and we really appreciate your suggestions. We'll revisit the layout and clarify the description to make things smoother, such as introducing the Outcomes earlier and avoiding the need to jump back and forth. Combining Sections 3 and 4 under one section is a great idea. Thank you again for these suggestions; they'll definitely help us make the paper better!
>
> * A random baseline achieves only about 19.11% accuracy (Table 6), and our system's better performance suggests that leveraging historical cases through probabilistic factor profiles and analogical reasoning is a meaningful approach for decision-making in unseen scenarios.
>
> ***
> If we've addressed your main concerns, could you consider raising your overall rating? We really appreciate the thought you've put into reviewing our work, and we'd love to keep the conversation going with any additional feedback!

---

> > ### Comment · Reviewer_2pMM · 2024-11-25
> > **.**
> >
> > Thanks! I am not really sure how I missed the context for Table 6, thanks for pointing that out.
> >
> > I still think the paper is a bit too dense, but authors can fix this in the final version.
> >
> > I appreciate this work and the general response is still good, I am updating my scores for both **contribution** and **rating**.

---

### Official Review · Reviewer_T5wU · 2024-11-02

**Soundness:** 3
**Presentation:** 3
**Contribution:** 3
**Rating:** 6
**Confidence:** 3

**Summary:**

This paper introduces DEFINE, a framework for enhancing LLM decision-making capabilities in complex scenarios, particularly focused on financial analysis of earnings call transcripts. The key contributions include:
1.A novel framework that constructs probabilistic factor profiles from complex scenarios and integrates them with analogical reasoning
2.A method for quantifying uncertainty in earnings call transcripts using key factors across macroeconomic, company-specific, and historical financial metrics
3.Implementation of the Bradley-Terry model to identify dominant factors and evaluate their collective impact on decision-making
4.Empirical validation showing improved performance over baseline methods in stock movement prediction tasks

**Strengths:**

1. this paper proposes a way to Successfully combines probabilistic factor profiles with analogical reasoning in a novel way and applied it in real world financial decision-making application.
2.The proposed method are evaluated in detailed ablation studies and verify the effectiveness.

**Weaknesses:**

1.The evaluation mainly focus on the financial domain and more cross-domain would strengthen the paper’s proposed method and claims.
2.Better and clear writing on some sections such as the Bradley-Terry Model.

**Questions:**

1.The 15 factors seems to be quite domain dependent and reply on human experts to select them. How robust does the proposed method against the forecasting factors.
2.In section 2.2, where is w_{xy} and w_{yx} defined? It’s not clear why use EM here, it would be better to add the EM details in the appendix.
3.The analogous reasoning method extract some similar history as context which is similar to RAG, can we add a RAG baseline for comparison?
4.Is there any ablation result without analogical reasoning?

---

> ### Author Response · Authors · 2024-11-23
> **Grateful for Feedback from Reviewer T5wU**
>
> __W1__
>
> Thank you for your insightful feedback! We agree that expanding the evaluation beyond the financial domain could further show the robustness of our proposed method. We're collaborating with researchers at a leading medical school to extend DeFine to medical decision-making under uncertainty. For example, DeFine identifies analogous patient cases using probabilistic factor profiles, helping physicians to address complex scenarios with greater confidence.
>
> __Q1-4__
>
> * Great observation! In our work, factors aren't hand-picked by experts but derived by querying LLMs about key variables in financial forecasting. This makes our approach adaptable and less reliant on domain expertise.
>
> * w_{xy} and w_{yx}​ represent how often Player x wins over y and vice versa. We use an EM algorithm here to iteratively refine our estimates of factor importance. Including EM details in the appendix is a great idea!
>
> * We're excited to explore a RAG Baseline! That said, RAG relies on embeddings or keyword matching to retrieve documents, which may not align well with our domain. The transcripts often use varied language to describe similar concepts. This variability can challenge RAG's retrieval approach, but we'd love to experiment with it.
>
> * Results without analogical reasoning are already included in Table 6.
>
> We really hope our responses have helped answer any questions you had about our work. If you feel our paper has improved as a result, could you consider raising your overall score? We're so grateful for the time you've put into reviewing this paper. Thank you so much; it means a lot to us!

---

### Official Review · Reviewer_LTXG · 2024-11-03

**Soundness:** 2
**Presentation:** 2
**Contribution:** 2
**Rating:** 3
**Confidence:** 3

**Summary:**

The paper introduces a framework for making stock predictions based on corporate earnings call transcripts called DeFine. It uses an approach that mixes LLM + bayesian decision making: it extracts domain-specific features (factor profiles e.g. "Regulatory changes" or "Political events" that may affect the future stock price) along with the probabilities of a given outcome for each feature using an LLM. The features are suggested by the LLM and filtered by the researchers. It then trains a model that, through paired comparisons, identifies the importance of each feature, to finally create a prediction model. The paper also introduces a dataset of corporate earning calls transcripts that may be useful for future research.

**Strengths:**

The paper introduces a new dataset for predicting whether to buy a stock given a corporate earnings call, carefully splitting by date, which may be useful for future research.

Its method is also a good example of how to combine LLM +

**Weaknesses:**

- The framing of the paper could be revised and rescoped: it is presented as a general analogical decision making framework, but in its current form it is only tested for stock prediction, and the feature selection seem specific to the domain. At times, I felt like this work would be better suited for a financial-specific venue. If the paper is reframed to be solely focused on the financial domain, it would be interesting to ground the feature selection on that field's literature.

- **Some claims about LLMs should be softened and/or corrected.** "LLMs are ideal for decision-making due to their ability to reason over long contexts and identify critical factors" is not a statement that should be made lightly, as research in planning with LLMs is still nascent and often requiring to offset the actual planning to more reliable components (e.g. a solver, as in TravelPlanner (Xie et al., 2024); or a domain-specific solver as in AlphaGeometry (Trinh et al., 2024)). There has been extensive research on LLMs reasoning abilities or lack of thereof, e.g. about their lack of generalization to unseen (Dziri et al., 2023). "LLMs are designed to provide reasoning traces for LLM decisions; however, their explanations remain ambiguous": this leaves out the key detail that these explanations are not causal and are often inconsistent with the final decisions or classifications (e.g. Wang et al. 2023). This may affect seriously the reliability of the approach.

- There is some mention about the weaknesses of other methods during the intro ("The latter often require extensive sampling during inference, which tends to increase inference costs and potentially leads to latency issues.") but never directly compared for cost-effectiveness.

- Presentation could be improved to make the methods explanation more focused and clearly define what is the final definition of DeFine. Background on Bayesian Decision Making can be left as a paragraph, and tone could sometimes be adjusted to be a better fit for scientific claims (e.g. "a methodology that optimally integrates textual reasoning with quantitative analysis"). Please consider removing a footnote linking to the Wikipedia page for the definition of “earnings call”.

- Results presentation could sometimes be improved to be able to grasp takeaways, e.g. Figure 2's confusion matrices. Could you have a metric to synthesize these takeaways and move the full confusion matrix to the appendix?

_References_

* Jian Xie, Kai Zhang, Jiangjie Chen, Tinghui Zhu, Renze Lou, Yuandong Tian, Yanghua Xiao, Yu Su. TravelPlanner: A Benchmark for Real-World Planning with Language Agents.  NeurIPS 2024.
* Trieu H. Trinh, Yuhuai Wu, Quoc V. Le, He He & Thang Luong. Solving olympiad geometry without human demonstrations. Nature 2024.
* Nouha Dziri, Ximing Lu, Melanie Sclar, Xiang Lorraine Li, Liwei Jiang, Bill Yuchen Lin, Sean Welleck, Peter West, Chandra Bhagavatula, Ronan Le Bras, Jena Hwang, Soumya Sanyal, Xiang Ren, Allyson Ettinger, Zaid Harchaoui, Yejin Choi. Faith and fate: Limits of transformers on compositionality. NeurIPS 2023.
* Peifeng Wang, Zhengyang Wang, Zheng Li, Yifan Gao, Bing Yin, and Xiang Ren. SCOTT: Self-Consistent Chain-of-Thought Distillation. ACL 2023.

**Questions:**

- In Table 5 and 6, what was the variance in the detected salience? I see all outcomes shown being at around 0.03, but I understand that there may be a sharp decline that might not be shown.
- You mention when discussing Table 3 that “DEFINE performs best at ‘Strong Buy’ recommendations and faces challenges with ‘Strong Sell’ categories. This may be due to its reliance on earnings call transcripts". Isn't this true of all/most methods compared?
- How would you concretely adapt this to other domains? What adaptations would you need to make and what assumptions or prerequisites would this new application need to have?

---

> ### Author Response · Authors · 2024-11-23
> **Appreciating the Feedback from Reviewer LTXG and Our Clarification**
>
> __W1__
>
> Thank you for your thoughtful suggestion about the framing of our paper. Our current experiments use financial data, this choice was intentional to demonstrate the robustness of our DeFine framework in a high-stakes domain with complex data. However, DeFine is designed as a modular, domain-agnostic framework that integrates probabilistic factor profiling and analogical reasoning to assist LLM decision-making under uncertainty. It can adapt to various contexts such as medical diagnostics. We will explicitly include examples beyond finance in our revised framing to show this adaptability.
>
> __W2__
>
> You're absolutely right that planning with LLMs is still in its early days, and frameworks such as TravelPlanner and AlphaGeometry are great examples of combining LLMs with external solvers to optimize over task-specific constraints. We'll definitely cite these works to give proper credit. DeFine can potentially complement this kind of planning research: we seek to improve LLMs' generalization to unseen scenarios using analogical reasoning. Instead of relying solely on external solvers, our method leverages historical cases to inform decisions in novel contexts. We see it as a piece of the larger puzzle for robust LLM planning.
>
> Our probabilistic factor profiles and analogical reasoning pipeline are separate from the reasoning traces LLMs generate (e.g., DeFine doesn't rely on those explanations at all). We'll make sure to clarify this further in the paper.
>
> __W3__
>
> Thank you for pointing this out! Let me clarify a bit why we didn't explicitly compare cost-effectiveness with other methods. Repeated sampling can be computationally expensive, especially when each sample requires an LLM API call with a long prompt. This wasn't feasible for us due to budget constraints, and it's also known that such approaches significantly increase inference costs and latency (Brown et al., 2024). DeFine avoids this by combining probabilistic factor profiles and analogical reasoning, which minimizes token consumption. This makes it inherently more cost-effective.
>
> (Brown et al., 2024) Bradley Brown, Jordan Juravsky, Ryan Ehrlich, Ronald Clark, Quoc V. Le, Christopher Re, and Azalia Mirhoseini. Large Language Monkeys: Scaling Inference Compute with Repeated Sampling. https://arxiv.org/pdf/2407.21787
>
> __W4-5__
>
> Thanks so much for these thoughtful suggestions! They're super helpful and will definitely improve our paper's presentation.
>
> We'll remove the footnote linking to Wikipedia for 'earnings call'. We'll also tighten up the Bayesian decision-making section to a concise paragraph, as you suggested, and adjust the tone of statements.
>
> You're totally right that summarizing the key takeaways with a metric would make the results easier to read. We'll add a synthesized performance metric to highlight those insights and move the detailed confusion matrices to the appendix.
>
> __Q1-3__
>
> * Regarding Tables 5 and 6, we'll consider including the variance in the salience values to give a clearer picture of how the factors distribute.
>
> * Great point! You're absolutely right. This challenge is common across most methods, as they originates from the biases often present in earnings call transcripts. Still, DeFine outperforms the baselines in mitigating this issue.
>
> * Thank you for this insightful question! We're actually collaborating with researchers at a leading medical school to explore how DeFine can be adapted for medical diagnostics. For example, a patient's medical condition is represented as a probabilistic factor profile, which evolves over time as symptoms change or treatments are applied. Similar to our work with earnings calls, we use analogical reasoning to identify comparable cases (patients with similar profiles) to help physicians make informed decisions in unseen situations.
>
> ***
> We've carefully addressed the points you raised. If you feel our efforts have meaningfully strengthened the paper, could you consider raising your overall rating? Thank you so much again for helping us make this a stronger paper; it means a lot to us!

---

> ### Author Response · Authors · 2024-11-25
> **Appreciating the Feedback from Reviewer LTXG and Our Clarification (Following Up)**
>
> Dear Reviewer LTXG,
>
> Thank you so much for recognizing our dataset contributions! Could you let us know if anything else needs clarification?
>
> We noticed your rating was significantly different from the other four reviewers, and we'd love to understand your viewpoint. If the updates improve our work, we'd be incredibly grateful if you could update your rating. Your help means the world to us!

---

> > ### Comment · Reviewer_LTXG · 2024-11-26
> > **Thanks for the clarifications & follow-up to the authors' response**
> >
> > Thank you for the clarifications!
> >
> > I still believe that it is difficult to support the claim of a framework being domain agnostic when it has only been tested in a single one. For example, it is hard to anticipate if the method would be as successful, as usually there are domain nuances that are difficult to foresee (e.g. just to mention one example, how effective would the LLM factor proposal be in other domains? how would an ineffective human filtering or a worse factor profiling affect the overall pipeline?). I am excited to hear that you are collaborating to apply this method to the medical domain in the future!
> >
> > Similarly, I am concerned with the way that LLM planning skills are currently portrayed in the manuscript (e.g. "LLMs are ideal for decision-making due to their ability to reason over long contexts and identify critical factors" is just one example), that goes beyond just adding citations to specific papers: I was trying to make the point that people are having to often rely on external solvers due to LLMs' weak planning skills, which contradicts the general tone of the intro. This method is focusing on high-stake domains, and uses LLMs as part of the methodology, hence why I find this especially concerning. I may have misunderstood something, but isn't the factor profile built using LLMs? Section 2.1 says the factors themselves are proposed by an LLM (with human filtering), and then lines 165-172 explain how the factor profile is built using an LLM. This would make your whole method rely on LLMs' planning skills even if you don't use the reasoning traces.
> >
> > Re W3: I understand LLM calls can get quite expensive, but I expected there would be at least an cost estimation clarifying budget constraints, to support how beneficial is this approach in comparison (e.g. one order of magnitude less? two orders?).
> >
> > Re W5 & Q1: I was expecting that as part of the response period we would be able to see some of these changes already reflected, as they should not difficult to make (e.g. computing the addition of a synthesized metric, or variance quantification).
> >
> > I hope this helps to understand my concerns better.

---

> > > ### Author Response · Authors · 2024-11-29
> > > **Grateful for Reviewer LTXG's Feedback and 2nd Follow-Up**
> > >
> > > Dear Reviewer LTXG,
> > >
> > > We noticed that your rating differs significantly from your comments and from those of the other reviewers. With only 4 days left for discussion, we'd love to fully address any remaining questions you might have. If we've satisfactorily addressed your points, could you please update your rating?
> > >
> > > Thank you again! Hope your Thanksgiving was fantastic. Looking forward to hearing from you soon!

---

> ### Author Response · Authors · 2024-11-27
> **Appreciate the follow-up Feedback and Our Further Clarification**
>
> Dear Reviewer LTXG,
>
> We appreciate this opportunity to clarify key aspects of our paper and address any misunderstandings that may have arisen!
>
> Firstly, we acknowledge the challenges associated with applying the DeFine framework to various domains. However, our collaboration with medical experts indicates its adaptability is promising, and we're enthusiastic about its potential for effective application across different fields.
>
> We concur with your observation regarding LLMs' planning abilities. We'll tone down the language in our paper to more precisely describe their role, so that we don't overstate their decision-making capabilities!
>
> We wish to clarify that DeFine employs a modular design, using a coarse-to-fine strategy to select factors, starting with broad categories such as macroeconomic trends, company-specific dynamics, and historical metrics, and then drilling down to the key variables. This design substantially improves reliability (with task decomposition done by experts, not reliant on LLMs' planning) in high-stakes scenarios.
>
> Moreover, we agree that LLMs' planning abilities have limitations, particularly in complex decision-making tasks. Recent research [1, 2, 3] has shown progress in reasoning over lengthy contexts, but we acknowledge this does not fully resolve planning challenges. We will adjust the manuscript's tone to more accurately reflect this balance.
>
> Regarding the cost analysis, we've conducted a preliminary evaluation showing potential savings compared to traditional methods, with estimates indicating a reduction of approximately $0.13 per data instance when avoiding repeated 10 samplings. We'll move Figure 2 to the Appendix for better flow. Additionally, the variances for Tables 5 and 6 are 5.55e-06 and 4.26e-06, respectively.
>
> We really value your insights and have made substantial efforts to address your concerns. We believe all these changes will enhance the clarity of our work. Could you let us know if there are any further points you'd like us to consider?
>
> *References:*
> 1. NeedleBench: Can LLMs Do Retrieval and Reasoning in 1 Million Context Window?, Li, etc, 2024
> 2. DetectiveQA: Evaluating Long-Context Reasoning on Detective Novels, Xu, etc, 2024
> 3. ALR2: A Retrieve-then-Reason Framework for Long-context Question Answering, Li, etc, 2024

---

> > ### Comment · Reviewer_LTXG · 2024-12-02
> > **Response to authors**
> >
> > Thanks for your response, I'll expand my main concerns below.
> >
> > "[...] Additionally, the variances for Tables 5 and 6 are 5.55e-06 and 4.26e-06, respectively." **The variances seem to be concerningly low: if I understand correctly, all influential factors have a extremely similar salience (all around 0.03), which would defeat the main goal of Section 6.2 of automatically detecting the influential factors and outcomes**. Am I missing something here?
> >
> > Thank you for providing a cost estimate. "[...] estimates indicating a reduction of approximately $0.13 per data instance when avoiding repeated 10 samplings". Could you at least provide a cost estimation for both techniques? Having just the difference between the two numbers not effective as a metric of cost reduction. I believe that if one of the main advantages of this method is its cost reduction, then this aspect should be clearly quantified.
> >
> > I agree that DeFine is promising to be applied to other domains, and I'm excited to hear that this is already in the works, but **I need to restrict my review to the paper in its present form, with the currently available experiments**: I still believe that the paper should be rescoped to not be presented as a general analogical reasoning frameworks given the uncertainties of adapting to a new domain. This single domain analysis is not properly advertised, as for example it's not possible to know this from the abstract, and make only an educated guess from the intro. I also believe there are missed opportunities to better highlight your work, e.g. make concrete statements on the benefits on DeFine in the intro.
> >
> > I checked but there has been no revision of the paper. It would have been important to see a first stab at this radical change in framing of LLMs' planning abilities. Besides acknowledging that LLMs have limited planning abilities, discussing how this may impact the method's reliability, as it's advertised specifically for high-stakes domains. Just to provide another example, there could be hallucination issues when summarizing the conversations that would affect the whole method's validity.
> >
> > Thank you again for your response.

---

### Official Review · Reviewer_r7rT · 2024-11-10

**Soundness:** 2
**Presentation:** 3
**Contribution:** 2
**Rating:** 5
**Confidence:** 3

**Summary:**

This paper incorporates factor profiles and analogical reasoning with LLMs to perform decision making in financial scenarios. Experiments are conducted to demonstrate the effectiveness of the proposal in classification like decision making.

**Strengths:**

• Proposed factor profile as the basic construct to perform decision-making instead of full text.
•  Prescribed a way to incorporate LLM outputs into Bradley-Terry models for estimating relative strengths of items and for bayes reasoning and analogical reasoning.

**Weaknesses:**

• The specific factor profile might limit the generalizability of the based LLMs. How to define factor profiles might become a human labor heavy task like the old days' expert systems.
• The way on how to incorporate LLM output to Bradley-Terry models, bayes reasoning and analogical reasoning is tailored to specific a small number of decision outputs like exemplified decision making. It is not clear how this framework can be applied to open set decision making restricting the original power of LLMs.

**Questions:**

1. In the abstract, the spoken speech aspect is mentioned as the a key challenge. However, in the main content the spoken aspect of the transcripts are not being given additional attention again and address the spoken aspect explicitly. If there is little special regarding the spoken aspect, it should be fine to not say so in the abstract.
2. Any criteria to select the factors? Why 15 factors are chosen for the financial sector? For other applications, what should be operable rules to identify the right set of factors? What exactly is the "iterative process of querying the LLM for key variables crucial in forecasting stock movements"? How can it be generalized to other domains?
3. What is the truth power and the insight of the proposal? Is it just about the similarity measure capability of  open set of texts and relevancy prefix-postfix conditional generation capability in LLMs  that replace the original hand-crafted similarity and relevancy in Bradley-Terry models and analogical reasoning?  Please discuss more?

---

> ### Author Response · Authors · 2024-11-23
> **Appreciate the Insights from Reviewer r7rT**
>
> __W1__
>
> Defining factor profiles doesn't require extensive human labor. It's a lightweight process. Our DeFine framework automatically identifies key factors by querying the LLM, requiring minimal human input for prompt writing. According to (Eigner and Händler, 2024), effective decision-making needs only on a small number of critical determinants, not an overwhelming list of variables. This makes our approach scalable.
>
> (Eigner and Händler, 2024) Eva Eigner, Thorsten Händler. Determinants of LLM-assisted Decision-Making. https://arxiv.org/abs/2402.17385v1
>
> __W2__
>
> Great question! Our framework is adaptable for more open-ended decision-making. While we focused on categorical outputs for this study, as it's easier to measure effectiveness quantitatively, the DeFine framework is inherently flexible. Analogical reasoning is at the heart of what we're doing, leveraging insights from similar past experiences to tackle new scenarios. This can extend to open-set decisions by drawing insights from analogous examples. We're excited to explore open-set decision-making in future work. Thanks for raising such a forward-looking point!
>
> __Q1__
>
> You're absolutely right. Spoken transcripts bring unique challenges, and we should've emphasized that more consistently. These transcripts are often long, containing hedging, vagueness, and overly optimistic tones (especially in executive remarks). This makes the system prone to bias, such as skewing toward 'buy' decisions with traditional approaches. We really appreciate you pointing this out and agree that either the abstract or the main content needs better alignment. Thanks for helping us make this clearer!
>
> __Q2__
>
> We used a coarse-to-fine strategy to select the 15 factors, starting with broad categories such as macroeconomic trends, company-specific dynamics, and historical metrics, and then drilling down to the key variables. We iterative prompt the LLM to identify factors affecting stock movements during earnings calls. This process ensures we focus on the key factors of decision-making without overcomplicating the model. For other domains, this approach can definitely be adapted. The same method could be used in other fields by updating the prompts to match the domain.
>
> __Q3__
>
> Great question! The true power of our approach lies in how it combines probabilistic factor profiles with analogical reasoning to tackle complex decision-making. Here's the breakdown:
>
> * We introduce probabilistic factor profiles that summarize complex scenarios and combine them with analogical reasoning to apply insights from similar cases.
>
> * Our method captures uncertainty in earnings call transcripts by focusing on key macroeconomic, company-specific, and historical metrics, and helps address the inherent ambiguity in spoken data.
>
> * We leverage the Bradley-Terry model to identify the key factors and assess their combined impact on decisions.
>
> * Our empirical results show substantial performance gains over baseline methods in predicting stock movements, which proves the applicability of the framework.
>
> If we've managed to address your concerns,  we'd be so grateful if you could consider raising the overall score; it would mean the world to us! Your thoughtful feedback has been invaluable, and we've taken it to heart to improve our work!

---

> > ### Author Response · Authors · 2024-11-29
> > **Appreciate the Insights from Reviewer r7rT and 2nd Follow-Up**
> >
> > Dear Reviewer r7rT,
> >
> > Your feedback has been invaluable and we've taken it to heart to improve our work. With only 4 days left for discussion, we'd love to fully address any remaining questions you might have. If we've satisfactorily addressed your points, could you please update your rating?
> >
> > Thank you again, and we hope you had an amazing Thanksgiving!

---

> ### Author Response · Authors · 2024-11-25
> **Appreciate the Insights from Reviewer r7rT (Following Up)**
>
> Dear Reviewer r7rT,
>
> We've carefully addressed your points and would love your feedback! Could you let us know if anything else needs clarification? Your feedback means the world to us! If the updates improve our work, we'd be incredibly grateful if you could update your rating. Thank you so much for your support!

---

### Meta-Review · Area_Chair_r4iK · 2024-12-19

**Metareview:**

Overall this is a good paper that introduces a framework for transcript analysis.  The paper contains many good insights about the nature of transcript writing and provides a framework that helps structure decision making using transcripts.  The primary weakness of the paper is that it seems to have only been evaluated on transcripts form the financial sector and the method of analysis is not trivially generalizable.  The authors do show that they can make predictions on a different sector of the financial markets based on training from another sector.

To make this paper stronger, please provide better guidance on how to expand the results beyond the financial sector and provide an example.

**Additional Comments On Reviewer Discussion:**

Two reviewers are recommending “marginal accept” and one marginal even though they do bring up the same weaknesses as the reviewer who is steadfastly rating the paper a “3”.  The reviewer who rated the paper a "3" was the most active in the discussion phase going through two rounds of communication with the author.  This reviewer had initially rated their confidence as "2" but in the process of the rebuttal phase increased their rating to "3" for self-assessed confidence.  Their score remained unchanged.

Unfortunately there was no response to the authors rebuttal from the other reviewers. Either they were busy or had little interest in the fate of the paper.  I reminded them twice but no response.

I gave the paper a quick overread myself and note that it is only tested with financial data and I believe that despite a theoretical contribution about how to separate out uncertainty, a non-negligible amount of heuristic work is necessary to make other kinds of decisions beyond the buy sell decision studied, making it of potentially limited utility to the ICLR community.  I wish there had been more discussion but the domain specific limitation seems like a valid weakness.

---

### Decision · Program_Chairs · 2025-01-22

Reject